# Disparities in Breast Cancer Mortality Rates in Israel among Urban and Rural Women

**DOI:** 10.3390/ijerph192315785

**Published:** 2022-11-27

**Authors:** Ronit Pinchas-Mizrachi, Judith Jacobson Liptz, Beth G. Zalcman, Anat Romem

**Affiliations:** 1Jerusalem College of Technology, Tal Campus, Jerusalem 9548311, Israel; 2Independent Researcher, Jerusalem 9322401, Israel

**Keywords:** breast cancer, mortality rates, urban, rural, sociodemographics

## Abstract

Breast cancer is a leading cause of death. There are a number of risk factors for breast cancer mortality including parity, age, ethnicity, genetic history, and place of residence. This study examined the disparities in breast cancer-related mortality rates among women from urban areas compared to rural areas in Israel. This was a retrospective, follow-up study on mortality from breast cancer among 894,608 Israeli women born between the years of 1940 and 1960. Data was collected from the Israeli Central Bureau of Statistics, the Population Authority, the Education Ministry, and the Health Ministry. Over 80% of women lived in urban areas. A higher incidence of mortality from breast cancer in Israel was found among urban women compared to rural women (1047.8/100,000 compared to 837/100,000, respectively). Even after adjusting for sociodemographic variables, higher mortality rates were found among women from urban areas in Israel compared to women from rural areas in Israel. It is believed that environmental factors can partially explain the geographic variation of breast cancer incidence, and that breast cancer incidence is likely a complex interaction between genetic, environmental, and health factors.

## 1. Introduction

Breast cancer is the second leading cause of death worldwide and is the cancer with the second highest incidence [1]. There are several documented risk factors for breast cancer mortality including age, ethnicity [2,3,4] breast-feeding habits [5,6], socioeconomic status (SES) [7], and living in urban areas [8,9]. Mortality risk from breast cancer goes up significantly with age and is especially prevalent in the age group of 40+ [10].

### 1.1. Risk Factors

#### 1.1.1. Parity and Breastfeeding

Some studies have found that parity, or the number of times a woman has experienced childbirth, impacts breast cancer risk among women [11,12]. However, it is unclear if this association is related to age at first pregnancy. Some studies indicate an association between a young age at first pregnancy and lower incidence of breast cancer, while other studies found no association between age at first pregnancy and breast cancer incidence [13,14].

The relationship between breastfeeding and breast cancer incidence is inconclusive. Some previous studies showed an inverse relationship between number of children that the woman breastfed over her lifetime, and total time of breastfeeding during her lifetime and incidence of breast cancer. Additionally, breastfeeding has been found to be an important protective factor for breast cancer development [15], even if the biological mechanism explaining this relationship is not fully understood [16]. However, some studies did not find a relationship between breastfeeding and incidence of breast cancer [17].

#### 1.1.2. Ethnicity and Genetics

Disparities in morbidity from breast cancer were found between ethnic groups [18]. Incidence among African women is lower than among European women [12], but survival among European women is higher than African women, due to higher rates of mammography screening [19]. Similarly, higher rates of incidence of breast cancer were found among black women than white women in the United States [20].

Genetic factors also play an important part in breast cancer development in women. Mutation of the BRCA1 and BRCA2 genes has been linked to worldwide breast cancer incidence [21] and this mutation is prevalent among women of Ashkenazi Jewish descent [22]. Two and a half percent of all Ashkenazi Jewish women are carriers of these mutations [23]. As such, the cross between ethnicity and genetics is important in predicting breast cancer risk. 

#### 1.1.3. Socioeconomic Status

A meta-analysis found higher rates of mortality and morbidity among women who had higher SES. Higher rates of survival for sick women were also found among women with higher SES [24]. Similarly, a cohort study conducted in Israel found higher incidence and survival rates among women with high SES compared to lower SES [25]. The disparities in survival rates between women with high SES and low SES can be explained by the stage of cancer at diagnosis. Sick women with higher SES tend to be diagnosed earlier compared with women with lower SES [26].

#### 1.1.4. Urban-Rural Disparity

There have been a number of studies that examined the difference in rates of breast cancer between urban and rural areas. A study conducted in China found that breast cancer incidence was significantly higher in urban areas than rural areas, and this difference was attributed to environmental pollutants [9]. A meta-analysis with studies conducted in the United States, Europe, Canada, South Africa and Egypt found that women from rural areas have significantly higher odds to be diagnosed with breast cancer at a more advanced stage compared to women in urban areas [8]. Another study conducted in the United States found that urban counties had a higher incidence of breast cancer compared to more rural areas. However, this association was partially mediated by county socioeconomic status (SES) and the general practitioner density. Furthermore, the study indicated that ethnicity plays a central role in incidence of breast cancer [27]. 

The disparity in breast cancer incidence between rural and urban areas may be due to a number of reasons, some favoring urban areas while others favor rural areas. On the one hand, certain cancers have been found to be higher in areas with higher population density [28]. Urban pollution has been shown to cause genetic mutations in wildlife, and this can help to determine the effects on humans [29]. An epidemiological review by the Silent Spring Institute has shown that exposure during three different phases of breast development (in utero, adolescence and during breastfeeding) to the chemicals dichlorodiphenyltrichloroethane (DDT), dioxins, perfluoro octane-sulfonamide (PFOSA), and air pollution as well as certain gasoline components, raise the risk of breast cancer [30]. On the other hand, rural areas tend to offer less access to healthcare services, including mammography screenings, which reduce breast cancer mortality [8,31].

### 1.2. Breast Cancer in Israel

From a global perspective, in 2018, Israel ranked 26 among nations for breast cancer incidence and 69 for mortality [3,32]. Incidence rates of breast cancer, as well as mortality rates from breast cancer are lower among Arab women than among Jewish women. A study conducted by the Israel National Cancer Registry found that Jewish women in Israel, have the highest rate of in situ breast cancer, compared to other women in several Middle Eastern countries, as well as Israeli Arabs [1]. However, studies conducted during the early 2000s found a much higher increase in the rates of breast cancer among Arab women as compared to Jewish women, while the increase in cases among Jewish women stabilized [32,33,34].

Survival rates among Jewish women is higher than survival rates among Arab women (90% compared to 84%). Mortality from breast cancer among Jewish women has been on a downward trend between 1996 and 2019, while the mortality rate among Arab women has remained relatively stable [32].

Due to the relatively high prevalence of breast cancer mortality in Israel, as well as known sociodemographic and environmental risks factors, this study aims to investigate disparities in mortality by place of residence (urban versus rural), in relation to other sociodemographic factors.

### 1.3. Study Objectives

To examine the disparities in breast cancer-related mortality rates among women from urban areas compared to rural areas in Israel.

## 2. Materials and Methods

A retrospective, follow-up study on mortality from breast cancer among women with Israeli citizenship born between the years of 1940 and 1960 over 32 years between the dates of 1 January 1990 and 31 December 2020 was conducted using data collected for the purposes of this study by the Israeli Central Bureau of Statistics from the Population Authority, as well as the Education Ministry and the Health Ministry. Work with the collected data was conducted in the research room of the Israeli Central Bureau of Statistics.

### 2.1. Study Population

The study population included 894,608 women who were Israeli citizens born between the years 1940 and 1960, of whom 730,965 were women who lived in communities with more than 20,000 residents (81.7%), and 163,643 (18.3%) were from communities with less than 20,000 residents. 

### 2.2. Study Variables

The study variables included: city of residence (urban area—all settlements with 20,000 inhabitants or more including settlements with a rural character; rural area—all settlements with less than 2000 inhabitants even if they are not agricultural and are not rural in nature), age, marital status, number of children, ethnicity (Jewish or Arab), place of birth, parents’ place of birth and SES relative to city of residence. Variables were obtained from the Population Registry.

The place of birth and parents’ place of birth variables were combined into the following three categories: Israel, Asia-Africa, and Europe-America. These variables were grouped once again into a single variable in the following manner: If the individual immigrated to Israel from another country, she was categorized according to the country from which she emigrated. If the individual was born in Israel and her parents were born in Israel, she was categorized as born in Israel, and if one parent was born in Israel and the other emigrated from another country, she was categorized according to the birth country of the parent who immigrated to Israel. If one parent immigrated from Asia/Africa and the other from Europe/America, the individual was categorized according to the birthplace of the mother. 

The SES relative to place of residence variable was ranked between 1 and 10, as defined by the Israel State Comptroller. The variable was broken down according to the following distribution: low (1–3), medium (4–6) and high (7–10). Details about the number of years of study were received from the Education Registry, which is managed by the Central Bureau of Statistics, and is based upon education data from various sources including: educational institutions, administrative files, surveys, comptrollers, administrative data, and more. This variable was grouped into three levels of education: high (15+ years of education), medium (11–14 years of education), and low (up to 10 years of education).

The outcome variable for this study, mortality from breast cancer, and year of death data were obtained from the Health Ministry.

### 2.3. Statistical Analysis

First, the distribution of the following variables was analyzed: age, number of children, marital status, country of birth, ethnicity, SES relative to place of residence, number of years of education, highest degree obtained by the individual, and rates of mortality from breast cancer among 730,965 women living in urban areas and 163,643 women living in rural areas. 

A T-Test is used in situations with a dependent quantitative variable and an independent categorical variable (with two groups). Therefore, a T-Test was utilized to determine the statistical significance of the differences between the groups in relation to quantitative variable, such as age. A chi-square test of independence is used to determine the relationship between two categorical variables. Therefore, the Chi-squared analysis was used to determine the statistical significance of the differences between the groups in relation to categorical variables. Level of statistical significance of the discrepancies between the groups were calculated based on a Chi-squared analysis for the categorical variables and the T-test for the age variable. The percentage of missing data relative to all the variables excluding the education variables stood at less than one percent. However, the percentage of missing data relative to education level stood at 20.2%.

Due to the missing information related to the education variables a new SES variable was constructed by combining two variables: education and SES relative to place of residence. Therefore, high SES was set as 15 or more years of education for those for whom this data was available and an SES relative to place of residence of 8–10 for those whom the data were missing. Medium SES was set as 11–14 years of schooling for those for whom these data were available, and SES relative to place of residence of 6–7 for those for whom education data were missing. Low SES was set as up to 10 years of education for those for whom this information was available, and SES relative to place of residence of 1–5 for those for whom education data were missing. 

Second, disparities in mortality rates over the follow-up period per 100,000 population were evaluated by study variables: size of community of residence, ethnicity, number of children, country of birth, and SES relative to place of residence as a combined variable. We evaluated the effect size of the disparities between groups and the level of statistical significance of these disparities, the AHR (Adjusted Hazard Ratio), was calculated mortality from breast cancer after adjusting for age. The AHR was calculated by measuring survival calculations using the SPSS program (version 23, IBM, Chicago, IL, USA), and building statistical models accordingly.

Third, regression and adjusted Kaplan–Meier curves models were created to evaluate the relationship between size of community of residence and mortality from breast cancer. In the first model, the following variables were entered: size of community of residence, age, and number of children. In the second model, the following variables were entered: size of community of residence, age, number of children, ethnicity, and country of birth. In the third model, the following variables were entered: size of community of residence, age, number of children, ethnicity, country of birth, and SES level as a combined variable.

## 3. Results

### 3.1. Demographics

The study group consisted of 894,608 female Israeli citizens born between 1940 and 1960. Of the study population 730,965 women lived in an urban area (81.7%), and contributed 21,959,251 follow-up years, and 163,643 women lived in rural areas (18.3%) and contributed 4,988,692 follow-up years.

Table 1 describes the distribution of the study variables among urban and rural women. The average age at the beginning of the follow-up period was higher for women in urban areas (39.07) compared to women in rural areas (38.34) (*p* < 0.001). The average number of children per woman in urban areas was lower (2.42) than the average number of children per woman in rural areas (3.35) (*p* < 0.001). In addition, significant differences were found between the SES of the groups. Among women in urban areas, a smaller percentage lived in areas classified as low SES (17.9%), compared to women in rural areas (24.5%). On the other hand, among women in urban areas, a smaller percentage were from areas classified as having a high SES (43.1%), compared to women in rural areas (49.7%) (*p* < 0.001). Almost 25% of the women in urban areas had up to 10 years of education, compared with 32.2% of women in rural areas (*p* < 0.001).

The mortality rates from breast cancer of the follow-up period among women from urban areas was 1047.8 per 100,000 women. In contrast, the mortality rates from breast cancer over the follow-up period among women in rural areas stood at 843.3 per 100,000 women. Similarly, the mortality rates from breast cancer per 100,000 life-years among urban women stood at 33.24, whereas the mortality rate from breast cancer per 100,000 life-years among rural women stood at 26.42 (*p* < 0.001).

### 3.2. Mortality from Breast Cancer by Sociodemographic Variables

In analyzing the roles of size of community in predicting mortality from breast cancer, after adjusting for age, higher rates were found among urban women compared with rural women (Age Adjusted Hazard Ratio (AHR) = 1.198, 99% CI (1.111–1.291)). There was no significant relationship between the number of children and mortality from breast cancer (AHR = 0.993, 99% CI = 0.981, 1.005). Lower mortality rates from breast cancer were found among Jewish women compared to Arab women (AHR = 0.809, 99% CI = 0.728, 0.899).

After adjusting for age, higher mortality rates from breast cancer were found among women who themselves or their parents were born in Europe/America/Australia, compared with women who themselves or their parents were born in Asia/Africa (AHR = 1.101, 99% CI = 1.025, 1.182). However, no significant differences were found in mortality rates from breast cancer among women who themselves or their parents were born in Israel, compared with women who themselves or their parents were born in Asia/Africa (AHR = 1.028, 99% CI = 0.962, 1.099).

In analyzing the role of SES in predicting mortality from breast cancer after adjusting for age, higher mortality rates were found among women with medium SES as compared with high SES (AHR = 1.139, 99% CI = 1.067, 1.216). However, no significant differences in mortality from breast cancer were found between women with low SES and high SES (AHR = 0.970, 99% CI = 0.903, 1.042). The distribution of breast cancer mortality rates by all study variables are found in Table 2.

### 3.3. Multivariate Models of the Relationship between Urban or Rural Area and Mortality from Breast Cancer

When the variables age and number of children were entered into a Cox multivariate model (Table 3) to evaluate the relationship between size of area of residence and mortality from breast cancer, higher mortality rates were found among women who lived in urban areas than among women in rural areas (AHR = 1.195, 99%. CI = 1.107, 1.2289). In this model, no significant relationship was found between number of children and mortality from breast cancer (AHR = 0.997, 99% CI = 0.986, 1.010).

In Model 2, with the following variables: size of community of residence, age, number of children, country of birth of the individual or her parents, and ethnicity, higher rates of mortality from breast cancer were found among women who lived in urban areas (AHR = 1.207, 99% CI = 1.117, 1306) than among women from rural areas. In this model too, no significant relationship was found between number of children and mortality from breast cancer (AHR = 0.994, 99%CI = 0.980, 1007).

In this model, higher mortality rates from breast cancer were found among women who themselves or their parents were born in Europe/America/Australia (AHR = 1.061, 99% CI = 1.001, 1.136) and among women who were born in Israel (AHR = 1.188, 99% CI = 1.099, 1.284) compared with women who themselves or their parents were born in Asia/Africa. Additionally, lower rates of mortality from breast cancer were found among Arab women than among Jewish women (AHR = 0.789 99% CI = 0.700, 0.887).

In Model 3, with the variables age, number of children, country of birth, ethnicity, and SES, higher mortality rates from breast cancer were found among women from urban areas, compared with women from rural areas (AHR = 1.178, 99% CI = 1.089,1.275). In this model, an inverse relationship was found between number of children and mortality from breast cancer (AHR = 0.946, 99% CI = 0.930,0.962) (Figure 1).

Higher mortality rates from breast cancer were found among women who themselves or their parents were born in Europe/America/Australia compared with women who themselves or their parents were born in Asia/Africa (AHR = 1.228, 99% CI = 1.125, 1.340). On the other hand, there were no significant differences found in mortality rates from breast cancer between women who themselves or their parents were born in Israel, compared with women who themselves or their parents were born in Asia/Africa (AHR = 1.064, 99% CI = 0.990, 1.144).

Additionally, lower mortality rates from breast cancer were found among Arab women than among Jewish women (AHR = 0.827 99% CI = 0.731, 0.939). In analyzing the role of SES in predicting mortality from breast cancer in this model, high rates of mortality were found among women with medium SES as compared with women with high SES (AHR = 1.145, 99% CI = 1.071, 1.225). Nonetheless, no significant differences in mortality rates were found between women with low SES and women with SES (AHR = 0.998, 99% CI = 0.922, 1.081).

## 4. Discussion

A higher incidence of mortality from breast cancer in Israel was found among urban women compared to rural women (1047.8/100,000 compared to 837/100,000, respectively). The average age of urban Israeli women is older than that of rural Israeli women. Urban Israeli women have fewer children than rural women, and there is a greater percentage of urban women from Europe/America in the larger cities as compared to rural women. However, the relative proportion of Israeli-born urban women is lower as compared with Israeli-born rural women. The proportion of urban Arab women is lower than the proportion of rural Arab women. In addition, differences were found in education level and SES between urban women and rural women. 

Jewish women had higher mortality rates than Arab women, and higher mortality rates were found among women who they or their parents were born in Europe/America/Australia as compared with women from Asia/Africa. 

Similar to previous studies, our study showed lower mortality rates from breast cancer among women with greater parity than women with lower parity [4]. This can be related to breastfeeding and hormonal change among pregnant women and new mothers [35]. The relationship between reproductive risk factors and breast cancer etiology is complex and can depend on the age of the woman when the breast cancer detected and a temporary additional risk after pregnancy [35,36,37]. 

Our study also found that Jewish women, particularly those of European/American descent, had higher rates of breast cancer mortality than Arab women. This finding is similar to findings from other studies, which have documented the significantly higher risk of breast and ovarian cancer among Ashkenazi women due to a higher rate of the BRCA 1 and 2 mutations. These mutations indicate an increased lifetime risk of developing breast or ovarian cancer [22,38]. 

These findings are also similar to data collected by the Israeli Ministry of Health, where higher rates of mortality from breast cancer were found among Jewish women than Arab women [32]. Furthermore, this study’s follow-up period begins in 1990, when disparities of breast cancer incidence between Jews and Arabs was much higher [32,33,34]. This may be able to be explained partially by parity differences between Jews and Arabs and genetic factors. Arab Israeli women have historically higher parity rates compared to Jewish Israeli women [39]. Additionally, BRCA 1 and 2 mutations are relatively common among Ashkenazi Jewish women, who constitute a significant portion of Jewish Israelis. 

Unlike earlier studies [24], higher mortality rates were found among women with medium SES compared to women with both low and high SES after adjusting to study variables. It is important to consider the fact that mortality rates from breast cancer comprises both morbidity rates and survival rates. It is possible that women with low SES tend to breastfeed more than women with high SES to save on the costs of alternative feeding options, which has been shown to be a protective factor against breast cancer [15]. However, one can also explain this phenomenon via survival chances of sick women. Women with higher SES likely have higher performance of mammography screenings compared with women with low SES and therefore benefit from early detection, which raises their chances of survival. Additionally, it possible that women with high SES benefit from better and timelier treatment which also increases their chances of survival [24].

However, even after adjusting for age, number of children, country of birth, ethnicity, and SES, higher mortality rates were found among women from urban areas in Israel compared to women from rural areas in Israel, similar to studies conducted in other countries such as China and the United States [8,9]. It is believed that environmental factors can partially explain the geographic variation of breast cancer incidence, particularly between industrialized and non-industrialized areas. In fact, numerous studies have shown an association between environmental factors, such as ambient and light pollution, and breast cancer incidence [40,41,42]. 

Several studies in the United States found that residence near major roadways was associated with an increased risk of breast cancer [43,44,45]. Traffic is a major source of pollution. Therefore, if urban women are exposed to larger amounts of traffic compared to rural women, it would seem that this could explain their higher risk of breast cancer. 

In reality, the incidence of breast cancer is likely a complex interaction between genetic, environmental, and health factors. Therefore, even if there is a higher overall rate of breast cancer mortality in urban areas, an Ashkenazi woman from a rural area may still have a higher individual risk of breast cancer than non-Ashkenazi women in an urban area, due to her genetic background. 

## 5. Limitations

There are a number of limitations to this study that must be addressed, primarily regarding data to which we did not have access. First, we did not have access to the following data, which would have enabled a more detailed analysis: total number of pregnancies a woman had, the number of women who breastfed and for how long, the women’s BRCA 1 and 2 carrier status, the types of breast cancer, or the characteristics of the area of residence (other than its size). These variables would have likely provided us with a better understanding of the study findings. Second, we did not have the education data of the participants’ husbands. This could potentially affect the participant’s socioeconomic status, particularly among women from more traditional communities. The family status variable was based on the woman’s family status at the beginning of the study and did not take into consideration any changes. This could potentially affect the findings of the study. We also did not have data on women who emigrated from Israel and did not return by the end of the follow-up period and disparities in emigration rates between urban and rural areas may introduce selection bias into the study. However, we decided to include their contribution to the study life-years until the time that they left Israel. The data used in this study were administrative data, therefore, we were unable to verify the cause of death. 

## 6. Conclusions

Higher rates of mortality from breast cancer among women from urban areas raises the need for more research to understand the environmental risk factors for breast and other types of cancer. In addition, it is important to identify environmental risk factors in urban communities and to initiate activities which will prevent environmental pollution, such as encouraging the use of public transportation over private car usage. We must also encourage development of technological solutions to prevent exposure to environmental pollutants during stages of breast development and raise aware of this danger among lawmakers and the greater population as well.

## Figures and Tables

**Figure 1 ijerph-19-15785-f001:**
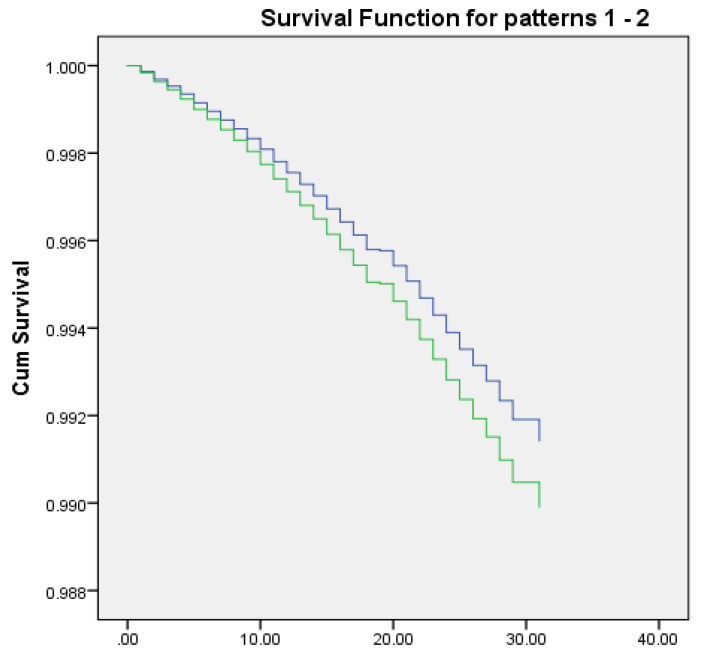
Analysis of Breast Cancer Mortality among Urban and Rural Women—adjusted for number of children, country of birth, subgroup in population, and SES (Model 3). Do you live in an urban area? no: 
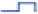
; yes: 
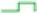
.

**Table 1 ijerph-19-15785-t001:** Study variables distribution among 730,965 urban women and 163,643 rural women.

		Urban Women	Rural Women	*p*-Exact Sig (2-Sided)	% Missing
		*N* = 730,965	*N* = 163,643		
Age	Mean (S. D.)	39.07 (5.70)	38.34 (5.71)	<0.001	0.00%
Number of children	Mean (S. D.)	2.42 (2.13)	3.35 (2.62)	<0.001	0.00%
Father’s Continent/Country of Birth	Asia/Africa	29.4%	22.6%	<0.001	0.00%
	Europe/America/Australia	45.4%	29.2%		
	Israel	25.2%	48.2%		
Subgroup in population	Jews	86.0%	69.9%	<0.001	0.00%
	Arabs and others	14.0%	30.1%		
SES by place of residence	Low 1–3	17.9%	24.5%	<0.001	0.88%
	Medium 4–6	39.0%	25.9%		
	High 7–10	43.1%	49.7%		
Highest Academic Degree	Middle school or less	22.1%	24.7%	<0.001	20.2%
	Matriculation exam or high school diploma	32.9%	29.7%		
	Postgraduate or Academic Degree	44.9%	45.7%		
Number of years of study	Up to 10	24.9%	32.2%	<0.001	20.2%
	11–14	43.5%	37.5%		
	15 or more	31.5%	30.3%		
Number of deaths from breast cancer during the follow up period		7669	1380		
Mortality rates from breast cancer per 100,000 women during follow up period		1047.8	843.3	<0.001	
Mortality rates from breast cancer per 100,000 life-years during follow up period		33.24	26.42	<0.001	

**Table 2 ijerph-19-15785-t002:** Distribution of breast cancer mortality rates by study variable: AHR (Age Adjusted Hazard Ratio) (*N* = 894,608 women).

		Mortality Rates from Breast Cancer during Follow Up Period Per 100,000 Women	Age Adjusted Hazard Ratio (AHR) (99% CI)
Total		1011.50	
Live in urban area	No	843.3	1.00
	Yes	1047.8	1.198 (1.111–1.291)
Number of children			0.993 (0.981–1.005)
Father’s Continent/Country of Birth	Asia/Africa	969.94	1.00
	Europe/America/Australia	1003.74	1.101 (1.025–1.182)
	Israel	1062.58	1.028 (0.962–1.099)
Subgroup in population	Jewish	1053.98	1.00
	Arab and other	803.20	0.809 (0.728–0.899)
SES by place of residence	Low	975.19	0.970 (0.903–1.042)
	Medium	1089.48	1.139 (1.067–1.216)
	High	1016.25	1.00

**Table 3 ijerph-19-15785-t003:** Results of Cox models predicting breast cancer mortality by demographic variable within the total study population (*N* = 894,608 women).

		Model 1	Model 2	Model 3
		HR (99%CI)	HR (99%CI)	HR (99%CI)
Live in Urban Area	No	1.00	1.00	1.00
	Yes	1.195 (1.107–1.289)	1.207 (1.117–1.306)	1.178 (1.089–1.275)
Age		1.063 (1.058–1.068)	1.062 (1.057–1.067)	1.063 (1.058–1.068)
Number of children		0.997 (0.986–1.010)	0.994 (0.980–1.007)	0.946 (0.930–0.962)
Father’s Continent/Country of Birth	Asia/Africa		1.00	1.00
	Europe/America/Australia		1.061 (1.001–1.136)	1.228 (1.125–1.340)
	Israel		1.188 (1.099–1.284)	1.064 (0.990–1.144)
Subgroup in population	Jewish		1.00	1.00
	Arab and Others		0.789 (0.700–0.887)	0.827 (0.731–0.936)
SES by place of residence	Low			0.998 (0.922–1.081)
	Medium			1.145 (1.071–1.225)
	High			1.00

## Data Availability

Data is available upon reasonable request.

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
