# Peer review of "Disparities in Breast Cancer Mortality Rates in Israel among Urban and Rural Women"

_ijerph, 2022, doi:10.3390/ijerph192315785_

Round 1
Reviewer 1 Report
In this paper, the authors present a retrospective study, which followed 984,608 Israeli women born between 1940 and 1960, aimed to examine the disparities in breast cancer-related mortality rates among women from urban areas compared to rural areas in Israel.
After a detailed review, I consider that this is an interesting study, suitable with the IJERPH journal. I have some doubts and recommendations for the authors. For that reason, I propose to do a Revision.
Please, see my comments in the attached pdf file.

Reviewer 2 Report
The authors have presented an interesting report on the important problem of addressing disparities in breast cancer mortality rates in Isarael among rural and urban women. I had a few minor comments:
1. Page 7 (Section 6.3): In the first paragraph, it seems like there is a typo in the adjusted hazard ratio reported for urban vs rural areas.
2. Table 3: Please correct the typo on the second row of Father's Continent/Country of Birth: "Europe/America/Australia
3. Table 3: What is the rationale for reporting 99% CI as opposed to the more commonly used 95% CI?
4. Figure 1 is missing a X-axis label.
Author Response
Dear Reviewer,
Thank you for the opportunity to revise our work. We have responded to the comments below, and we hope that you find our responses satisfactory. We greatly appreciate your time in reviewing our manuscript.
Kind regards,
The authors
- Page 7 (Section 6.3): In the first paragraph, it seems like there is a typo in the adjusted hazard ratio reported for urban vs rural areas.
Thank you for this comment. We have fixed this accordingly.
- Table 3: Please correct the typo on the second row of Father's Continent/Country of Birth: "Europe/America/Australia
Thank you for this comment. We have fixed this accordingly.
- Table 3: What is the rationale for reporting 99% CI as opposed to the more commonly used 95% CI?
Thank you for this comment. We chose to report 99% CI due to the large sample which enables reporting 99%CI. Additionally, the fact that the sample is so big means that there is a bigger chance to fund significant disparities among the groups at a level of significance that is lower than 95%.
- Figure 1 is missing a X-axis label.
Thank you for this comment. This graph is a survival curve, and the x axis, which always represents time in survival curve, is the number of years that have passed since the beginning of the follow-up period. The y axis represents the relative number of survivors. The beginning of the follow-up period starts with 1.00, which is the total population with which we started the follow-up, and later on there is a smaller proportion, based on the relative population that has survived and is still being followed up.
Round 2
Reviewer 1 Report
Thank you. I do not have more questions.